# Enthalpy-Sensing Microsystem Effective in Continuous Flow [note 1]

**DOI:** 10.3390/s19030566

**Published:** 2019-01-29

**Authors:** Taoufik Mhammedi, Lionel Camberlein, Frédéric Polet, Bruno Bêche, Etienne Gaviot

**Affiliations:** 1Laboratoire d’Acoustique de l’Université du Mans, LAUM, UMR CNRS 6613, 72085 Le Mans, France; Lionel.Camberlein@univ-lemans.fr (L.C.); frederic.polet@univ-lemans.fr (F.P.); etienne.gaviot@univ-lemans.fr (E.G.); 2Institut de Physique de Rennes, IPR, UMR CNRS 6251, 35042 Rennes, France; bruno.beche@univ-rennes1.fr; 3Institut Universitaire de France (IUF), 103 bd saint-Michel, 75005 Paris, France

**Keywords:** heat flux transducer, planar thermopile, enthalpy of mixing, conjugated variables, 3D-printed mixing chamber, SU8 channels, thermal differential and common modes

## Abstract

A new microsystem designed to detect and measure in real time the enthalpy of mixing of two fluid constituents is presented. A preliminary approach to quantify the enthalpy of dilution values or mixing is first discussed. Then, a coherent rationale leading to structure devices operating in real time is formulated, considering the straightforward assessment of heat-flux transducers (HFTs) capability. Basic thermodynamic observations regarding the analogy between thermal and electrical systems are highlighted prior consideration of practical examples involving mixing water and alcohols. Fundamentals about HFT design are highlighted before presenting an adequate way to integrate both functions of mixing and measuring the entailed heat exchange as two continuously flowing fluids interact with one another. Thereby, the development of a relevant prototype of such a dedicated microsystem is discussed. Its design, fabrication and implementation under real operating conditions are presented together with its assessed performance and limits so as to highlight the advantages and shortcomings of the concept.

## 1. Introduction

As regards industrial processes, assessing the heat exchanges occurring as two fluid constituents are mixed together is a most common requirement, whatever the quantities to be considered. Furthermore, miniaturized systems bring, nowadays, new challenges especially in the case of constituents interacting in continuous flows. As a state variable and the intensive quantity associated with thermal energy, the absolute temperature is classically the only prominent quantity to be assessed by way of calorimetry. Then, thermal metrology hinges on thermometry and its numerous methods [1]. However, the extensive quantity conjugated with temperature is the entropy current that should be of most valuable interest insofar as dedicated sensors would be available [2]. Indeed temperature and entropy are conjugated quantities with regard to thermal energy, in the same way that voltage and electric charge are the conjugated quantities characterizing any electrical exchange [3,4]. 

Given any domain of energy, the comprehensive knowledge of a system is allowed as both conjugated quantities are within metrological reach [1,5]. Then, considering thermal exchanges, on a metrological standpoint, when small changes of temperature are involved, heat-flow meters (HFMs) may supply quite relevant intelligence especially as regard detection of weak thermal changes [6,7]. For instance, considering phase transitions with quasi-isothermal plateaus, the HFM approach has already proved its substantial metrological advantages [8,9]. Beyond such an approach, in this paper we highlight a valuable way to take advantage of the planar thermopile constitutive of a generic HFM so as to design a sensitive area fitted for measuring weak thermal exchanges otherwise hardly detectable in terms of ratio signal/noise.

As miniaturized devices, we first describe dedicated HFM-based structures before detailing the aforementioned original design featuring a specific planar thermopile overlaid with micro-SU-8 channels [10] arranged on a glass substrate. Besides presenting the advantages of self-generating sensors (no power supply, 2 wires), such alloy-based thermopile configurations feature low output impedance values (Z_0_ < 2 kΩ) and naturally minimise Johnson electrical noise.

Since several designs have been investigated, the various stages in the development of the most miniaturized prototype (microsystem) are delineated. Then, as practical examples of specifications and verified performance, relevant experimental processes involving mixing water + ethanol and water + propan-2-ol have been investigated [11,12].

## 2. Materials and Methods

Considering calibration, specifications for designing effective devices rely on data available in the literature, especially as regards fluid constituents to be mixed. Moreover, as a mandatory requisite the constituents experimented with should be devoid of any kind of interaction with the materials constituting the sensors. Prototypes involving depron (λ = 0.027 W/mK), polylactic acid (PLA: λ = 0.13 W/mK), polyethylene terephthalate (PET: λ = 0.14 W/mK), SU8 (λ = 0.15 W/mK), and glass (λ = 1 W/mK), first appeared rugged enough with mixings of water with alcohols. However, PLA was afterward replaced with PET, both materials being easily shaped with a 3D printer.

### 2.1. Prerequisite for Calibration: Suitable Fluids and Ways to Handle their Mixing

Both ethanol and isopropyl alcohol (IPA) mixed with water have been investigated, considering references 11 and 12. Products used in this paper were high-quality products (Ethanol Rectapur™ 99.5% in vol. and IPA Normapur™ 99.7%) mixed with deionised water (10.6 MΩ·cm). Taking advantage of both the Boyne’s study [11] and the Peeters and Huyskens polynomial model [12], the excess enthalpies of mixing for any case of combination may be calculated. For instance, one may resort to the following fitting equation, where X_W_ and X_A_ denote the respective mole fraction of water and alcohol within the blend experimented with:(1)$$\Delta H_{\mathrm{mix}}=\;C_{6/1}\cdot X_{w}^{6}\cdot X_{A}+\;C_{1/1}\cdot X_{w}^{}\cdot X_{A}+\;C_{1/2}\cdot X_{w}^{}\cdot X_{A}^{2},\;\mathrm{with}\;XW_{\;}=\;(1-X_{A}),$$

As depicted in Figure 1, such a thermodynamic behaviour may be illustrated whatever the quantitative combination involving water and both alcohols. 

Considering a given mass of water M added with that of an alcohol m and referring to the entailed ratio r = m/M, the mole fraction of alcohol may be written as:(2)$$X_{A}=\;\frac{r\cdot w}{1+r\cdot w}\;.\;\mathrm{Here},\;w=\;\frac{MW_{water}}{MW_{alc}}\;\mathrm{is}\;\mathrm{the}\;\mathrm{ratio}\;\mathrm{of}\;\mathrm{respective}\;\mathrm{molecular}\;\mathrm{weights},$$

Practical values are: {MW_water_ = 18.015 g/mol, MW_IPA_ = 60.1 g/mol, MW_Ethanol_ = 46.07 g/mol}. Then, considering energy balance, the effective exchanged heat ΔQ (J) that can be measured by way of standard calorimetric methods may be given with:(3)$$\Delta Q=\;\Delta H_{\mathrm{mix}}\cdot \left[\frac{m}{MW_{alc}}+\frac{M}{MW_{water}}\right]\;$$

Considering now a continuous flow process involving both constituents, the entailed heat flow Φ (W) produced in steady state and related with Equation (3) may be written as:(4)$$\Phi =\frac{\mathrm{dQ}}{\mathrm{dt}}=\;\Delta H_{\mathrm{mix}}\cdot \left[\frac{\rho_{alc}\cdot \left(\partial V_{alc}/\partial t\right)}{MW_{alc}}+\frac{\rho_{water}\cdot \left(\partial V_{water}/\partial t\right)}{MW_{water}}\right]\;$$

It is clear that as peristaltic pumps were used to introduce fluids within the prototypes, m and M values had to be inferred from their controlled respective volumetric quantities with ρ__IPA_ = 0.785 g/cm^3^ and ρ__Ethanol_ = 0.789 g/cm^3^.

### 2.2. Generic Heat Flow Meter (HFM)-Based Design for Monitoring Excess Enthalpies

Heat flow meters (HFMs) are nowadays quite common self-generating sensors allowing measurement of heat transfers with internal low noise due to their reduced output impedance. Indeed, Z_0_ may range between 100 Ω and 2 kΩ [1-chap.XI, 6,7,8] with the most common devices designed in the laboratory. A generic structure, based on a planar thermopile, is depicted in Figure 2a. Basically, such a device relies on differential thermocouples arranged in series along a meandering path. Then, as one thermojunction out of two is subject to the heat flow driven through the upper collector, the numerous entailed local voltages are added in series accounting then for the thermal energy proceeding across the sensing area. The top view of a real specific set of plated thermoelements is given with Figure 2b. Practically, sensitive areas may be realized with sizes ranging from 1 cm² up to 25 cm² depending on either microtechnologies with sputtering processes, or classical printed circuit techniques involving electroplating. Both approaches hinge on photolithography [13] and the manufacturing process is pliant enough to customize planar thermopiles on request, even fitted for radiant measurements [7,14]. With response time values in the order of one second, HFMs, functional in the field of control, come up as noteworthy solutions in case of unsteady transfer analysis.

As regards the requirements dealt within this paper, we investigated continuous mixing processes with a first family of miniaturized devices. They were fabricated with standard HFMs assembled with PLA and PET 3D-printed-mixing-chambers. In such a configuration the mixing chamber (whose bottom face must be thermally insulated) was arranged on the lower face of a HFM whose upper face was superimposed with a heat sink. The generic layout of such a prototype is depicted in Figure 3a. Then, a real structure is illustrated in Figure 3b.

In any case, an effective isothermal condition must be imposed on both liquids before being mixed. Making them circulate within separated channels arranged within the upper sink was thereafter considered as an advantageous solution enhancing then the design of Figure 3.

Considering a given HFM with a planar thermopile structure involving N couples of thermojunctions, the amplified Seebeck output voltage is representative of the heat current [Φ_mes_ (W)] proceeding through the acting surface with:(5)$$V_{\mathrm{out}}=\;G\cdot \mathrm{Rs}\cdot \Phi_{\mathrm{mes}}\;=\;G\cdot N\cdot \Delta \alpha \cdot \delta \theta \;,$$
where G is the gain of the amplifier (50 < G < 1000), Rs the responsivity of the HFM (V/W), Δα the relative Seebeck coefficient (Δα ≅ 38 µV/K) and δθ the local elementary differences in temperature available between two adjacent thermojunctions as depicted in Figure 2a. Although quite liable, such systems based on HFMs featured a major drawback due to internal turbulence within the mixing chamber: then, a significant enhancement regarding the operating principle had to be considered.

### 2.3. Specific Design Relying on Micro-Technology

Any extraneous noise due to uncontrolled turbulence comes up as an unfortunate drawback while considering most common control processes: generally, the volumetric flow of a solute has to be adjusted for a given flow of the solvent, the enthalpy of mixing being determined with prior assessments (either theoretical or experimental). The sensor presented in this section has been designed with a view to making possible the adequate control of the solvent flow as regards industrial applications. Then, we had to minimize the integrated effect of turbulence impeding the measurements over the sensing surface of the HFM. To overcome such an issue, we changed the whole HFM approach: indeed we made it possible to locally ensure the mixing process directly upon one thermojunction out of two of the thermopile as depicted in Figure 4, the other one being at the relative reference temperature—that of both constituents—in the course of the mixing process.

We may then consider the thermopile output Seebeck voltage with the right term of Equation (5) while highlighting the local differences in temperature (δθ): the relevant emf added values come up as representative of statistics on thermoelectric voltages localized on the numerous sets of thermojunctions; we verified that it could be considered as a valuable output signal practically associated with the mixing heat flow described with Equation (4). Then, regarding a given mixing process in a steady state with an expected theoretical heat flow rate Φ for a fixed water flow set point value, we may consider after amplification:(6)$$V_{\mathrm{out}}=\;G\cdot \xi \cdot \Phi \;\;,$$
with G the gain and ξ (V/W) an apparatus-related constant depending on the design of the thermopile and its overlaying channels driving both constituents.

## 3. Results: Technological Achievement and Validation of the Microsystem

As depicted in Figure 5 the aforementioned thermopile is deposited on a glass substrate. Then, walls arranged together with apt studs and blades made of SU8 250 µm in thickness allow an adequate repartition of both constituents to be mixed together.

### 3.1. Micro-Fabrication

#### 3.1.1. Specifications and Features

The prototype experimented with includes a thin planar thermopile featuring 360 plated thermoelements, coated with 3 matched compartments over a whole 1.44 cm² acting surface (Figure 6a). The fabrication process is single sided, and requires six mask layers: 1 for a first SU8 layer, 2 for the bimetallic thermopile, 1 for ensuring both the electrical insulation and chemical protection, 1 for building the walls and pads and 1 for the upper face (covering lid). Indeed the device is mounted on a standard glass substrate 1 mm in thickness [Elka® Micro Slides 76 × 26 mm, λ = 1 W/(m·K)], allowing a fair reference isothermal condition. The thermopile relies on the constantan thermoelectric alloy [poorly conducting material; σ_1_ = 52 (µΩ·cm)^−1^] plated with gold conductors [highly conducting material; σ_2_ = 2.2 (µΩ·cm)^−1^] prior covering with the patterned SU-8 layers. It must be noted that numerous parameters are involved to account for the optimization of such a thermoelectric device; thermal and electrical conductivities, the effective thermoelectric power of the plated structure and dimensional parameters are among the most important to be considered [7]. For instance, the first SU8 layer arranged between the substrate and the constantan allows optimized δθ values between each thermojunction.

#### 3.1.2. Manufacturing Principle

At first, after chemical and plasma cleaning, the glass substrate is coated with the aforementioned thermally insulating SU-8 layer (Microchem® 2100 mask #1), 100 µm in thickness (post-bake at 95 °C, λ = 0.3 W/(m·K)). A constantan layer (600 nm) is deposited by RF magnetron sputtering (PLASSYS-BESTEK, Evry, France MP-450-S). Then, a continuous meandering path (50 µm in width, legs 50 µm apart: mask #2), ended with the contact pads, is wet etched, prior to deposit the gold-plated electrodes. The latter (400 nm in thickness, 2 mm in length, 50 µm in width: mask #3) are processed by classical lift-off techniques [13]. Then, a SU-8 protective and electrical insulating layer (10 µm; MicroChem^®^ Corp, Westborough, MA, USA 2010) is spread, cured, and recessed by photoimaging (mask #4). Thereafter, the walls, blades and surrounding enclosure (Figure 6a) are patterned by way of a thick SU-8 layer (250 µm: MicroChem^®^), with the second last photolithographic stage (mask #5). Eventually, the device is oxygen-plasma treated so as to optimize the surface wetting properties, and the thin holes are drilled by way of sand blasting through the glass substrate allowing the water-based fluids to be steadily distributed into the set of channels both through capillary action and pressure due to the peristaltic pumps. Eventually, the lid (fitted with the outlet channeling) allowing the closing of the acting volume is fabricated with same techniques.

### 3.2. Microprototype Characterization

According to the allotted dimensional parameters, especially the effective area of the glass substrate, and taking account of the characteristics of the peristaltic pumps, preliminary tests were operated as follows:

#### 3.2.1. Preliminary Tests with Classical Temperature Measurements

A first set of experiments has been carried out to evaluate under realistic conditions the relevance of Equations 1 and 3. To this end, given masses (weighted ± 0.02 g) were mixed within a 20 cm^3^ thermally insulated cell with classical calorimetric procedures. Temperature values were measured with a G10K3976 radial glass thermistor (NTC) fitted with a thin fin (aluminized aluminium foil # 1 cm²), operated with a 100 µA current value (HP 33120A). After a straightforward determination of the threefold set of Steinhart–Hart coefficients [15], temperature values were measured with an accuracy estimated to the nearest 0.05 °C within the range {10 °C–60 °C}. Since our characterizations regarding alcohols and water were in fair agreement (albeit less accurate due to our oversimplified cell) with that of Peeters and Huyskens investigating with a Parr Calorimeter [12], and with no further interest than that of classical thermometry, they may be considered of no interest in this paper whose topic is focused on applicability of heat flow measurements. It may be noticed that as a prospective source of error we had to eliminate any PLA layer within the cell due to its slight chemical interaction with both water and alcohols. Then, we verified (haphazardly) that due to its complex structure water may dissolve almost anything in the long run [16].

#### 3.2.2. Suitable Conditions to Operate the Microsystem

As a consequence of the dimensioning of the prototype, relevant working regimes had to be identified. To this end, a first coarse setting with water highlighted proper flow values ranging around 7 mL/min. Then, both inlets were supplied with coloured water (with, respectively, blue and yellow food dyes): a straightforward eye sighting of the green coloured volume allowed us to choose an apt duty point for the water flow (water as the solvent) and delineate the adequate range for the solute (alcohols). Hence, the best suited controlled water flow was fixed at 7.2 ± 0.2 mL/min. At this stage, first tests highlighted the critical issue as regards removing entrapped air bubbles as is clearly shown in Figure 7a. As regards filling tests, the detail of the green expanding area in a given transient regime is shown in Figure 7b.

Figure 8a illustrates the global set up involving the system under experiment together with the two peristaltic pumps. Blended fluids were directly evacuated in a simple beaker. The maintenance of the atmospheric pressure within both burettes containing the liquids to be mixed was ensured by way of unscrewing their respective cap. The output voltage of the sensor was amplified (Chopped TI^®^ Texas Instrument, DALLAS, USA-TLC 2652A: G = 1000) and filtered (2d order, LP-3dB: 10 Hz) before signal acquisition. The output signals volumetric flow values obtained with IPA and ethanol are illustrated in Figure 8b.

## 4. Discussion

### 4.1. Quantitative Results Regarding the Microsystem: Presentation and Relevance

Considering Figure 8b, theoretical values are quantitatively inferred from Equation (4), regarding IPA and Ethanol associated with the water duty point set at 7.2 mL/min. The entailed expected behaviour is shown in solid lines with ΔH_mix_ considered for t °C = 25 °C. On the other hand, experimental results are depicted by way of specific boxes accounting for uncertainties regarding both flow values and related monitored output voltages. With such a configuration the ξ parameter binding the thermopile emf and the theoretical flow (Equation 6) can be established as: ξ = (2.083 ± 0.02) × 10^−4^ V/W. However, we must underline a slight ξ-dependence on the water flow set point values. It must also be noted that experimenting with ethanol brought (by a far cry) more difficulties than with IPA. Then, the effective operating range with ethanol was substantially reduced due to incontrollable internal degassing mechanisms that were involved (although barely visible to the naked eye), entailing extraneous output signals. Other prototypes with different sizes will be realized for a comprehensive analysis of the behaviour of the system.

### 4.2. Advantages, Shortcomings and Prospects

Although designed with a same view to monitoring heat exchanges in continuous flow, HFM-based sensors (Figure 2) and devices working with a distributed mixing chamber (Figure 4 and Figure 5) are systems quite different from one another. Since HFM configurations hinge on heat exchanges through the sensing surface, their output voltage depends on the temperature gradient perpendicular to the sensing surface: then, a heat sink is mandatory for proper operation. Conversely, composite mixing chambers allow a direct local heat distribution upon one thermojunction out of two; then, each heat exchanges entails its respective δθ (parallel with the plane of the thermopile) working as a thermal differential mode: moreover, insofar as the global heat is evacuated, the mean temperature of the substrate acts as a thermal common mode with a somewhat limited influence on the output signal due to the C_xy_ parameters drift (Figure 1) function of temperature (impacting ΔH_mix_ in Equation (1)). Hence, with small chemical interactions the heat sink is optional.

Moreover, such a distributed configuration features another significant advantage as regards the response time whose value is naturally quite reduced (<0.5 s) compared with that of more cumbersome HFM devices. Then, our preliminary results highlight a promising ability for such devices to be implemented within regulatory loops [17], insofar as their operating regimes are subjected to a comprehensive control. Although the relevance of the concept (together with its feasibility) has been proved, several critical issues remain to be addressed. Indeed, operational implementation may be fatally flawed in the case of air bubbles (even with a trifling amount, ethanol being more difficult than IPA to experiment with) introduced within the mixing chamber. Then, before operating the system has to be cleaned for any trace of air that must be drained out with a comprehensive flush of liquid. Unfortunately this may come up as a significant drawback as regards fluid waste. Furthermore, operating regimes hinge on numerous parameters. Beyond the obvious dimensional ones, the hydrophilic properties depend critically on the effectiveness of material processing, notably the SU8 plasma treatment (Plassys® CVD 300, oxygen Plasma). With initial trials, the effects were sometimes tricky, entailing erratic signals. Eventually, in operation another potential default may stem from the positioning of the device. We assume that the related observed drifts were due to the difference in density of the fluids experimented with.

## 5. Conclusions

New sensors designed to monitor in real time the heat exchanged while mixing two fluids have been developed. Based on planar thermopiles, two families are presented. The first one comes up as a direct application of heat-flux transducers: a 3D-printed realization of such a configuration as mini-system is delineated. The second family relies on a distributed mixing chamber and its relevant generic design is presented whatever the size of the device. However, with a view to addressing pharmacobiology applications, a specific development with microtechnologies is presented. After describing the fabrication of such a microsystem, its implementation method is discussed on the ground of relevant tests carried out under realistic conditions with water and alcohols (IPA and ethanol) mixings. Then, the advantages and shortcomings are pointed out. Finally, industrial applications involving the device within a regulatory loop controlling the solvent flow are expected in the coming year.

## Figures and Tables

**Figure 1 sensors-19-00566-f001:**
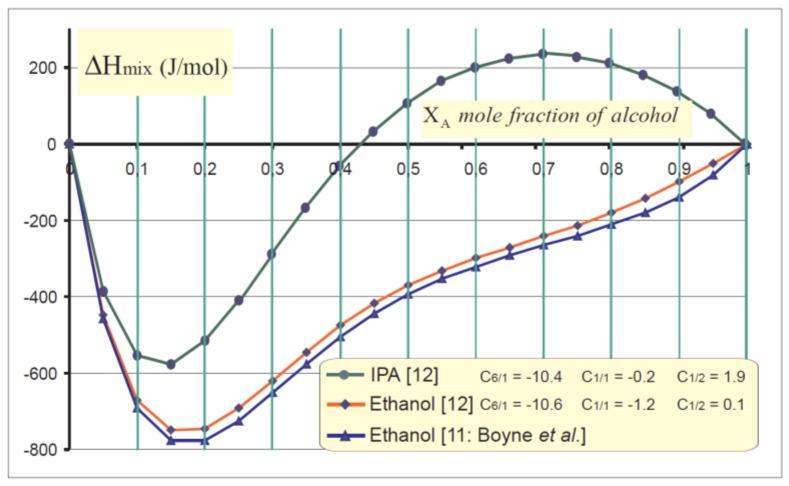
Heat of mixing of water with alcohol mixtures (J.mol^−1^) vs. mole fraction of alcohol (X_A_). It appears that as isopropyl alcohol (IPA) + water are blended together, processes can be either exothermic (ΔH_mix_ < 0) or endothermic.

**Figure 2 sensors-19-00566-f002:**
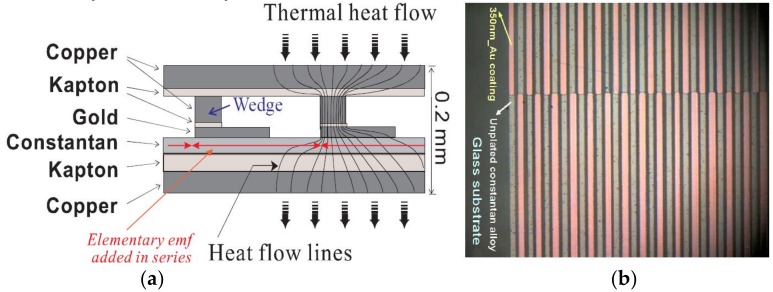
Heat flow meter (HFM) fundamentals: (**a**) cross section view of a generic structure: heat flow is constricted through a copper wedge so as to act on one gold-constantan contact out of two. (**b**) Arrangement in a checkerboard pattern for a specific golden plated planar thermopile deposited on a glass substrate.

**Figure 3 sensors-19-00566-f003:**
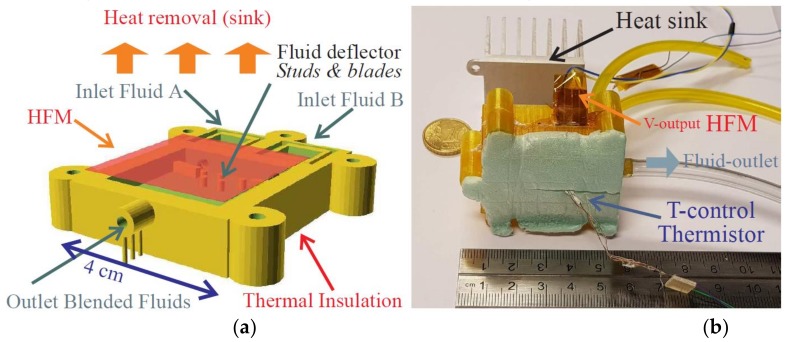
Generic HFM-based devices: (**a**) layout-design of the polyethylene terephthalate (PET) 3D-printed unit. Inside the mixing chamber, blades and studs are arranged so as to enhance the self-stirring of both fluids. (**b**) Photograph of a prototype apart from its heat sink. Isothermal channels drive both inlet fluids within the mixing chamber. Then any imbalance in heat entails an effective demand on the heat sink through the HFM.

**Figure 4 sensors-19-00566-f004:**
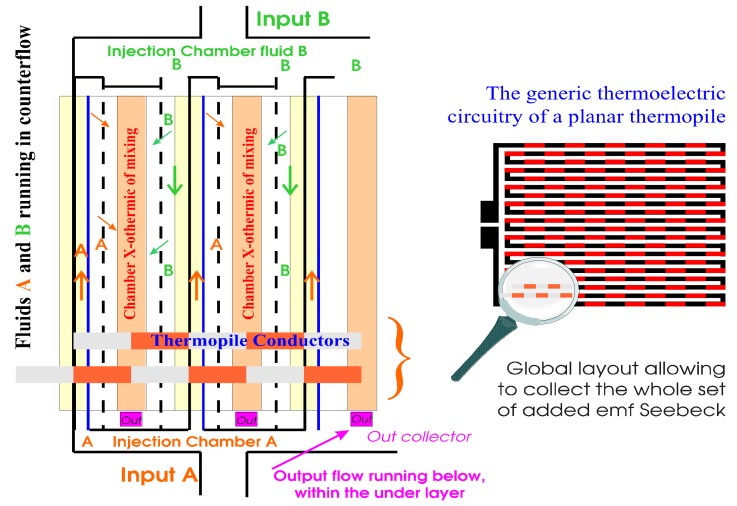
Main principles for constitutive parts of the self-generating microsystem (Zo < 2kΩ).

**Figure 5 sensors-19-00566-f005:**
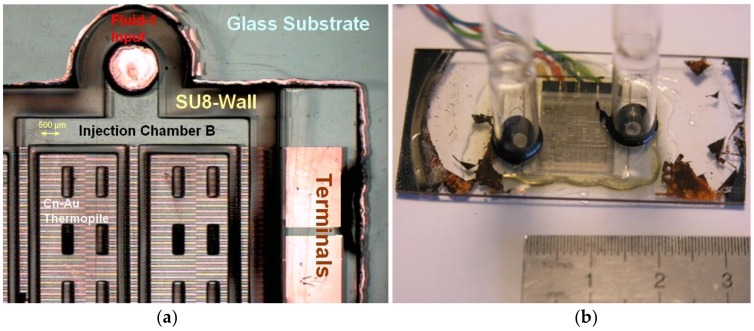
(**a**) Layout of the prototype highlighting: the glass substrate (λth = 1 W/m·K), the Au-Constantan plated planar thermopile (N = 360), the SU8 walls 400 µm in height allowing to drive and mix both fluid-constituents, and the electrical terminals. (**b**) Global view of the device with both fluids inputs: the output being on the other side is here invisible.

**Figure 6 sensors-19-00566-f006:**
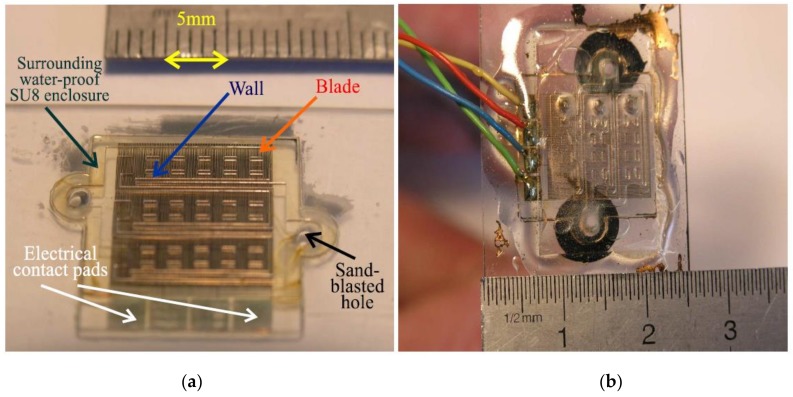
(**a**) Layout of the mixing chambers arranged upon the planar thermopile. (**b**) The device seen through before closure (its lid being fitted with the mixing outlet).

**Figure 7 sensors-19-00566-f007:**
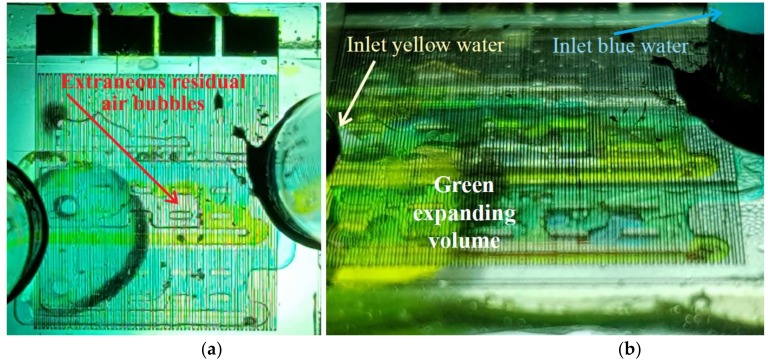
Filling tests: (**a**) air bubbles entrapped within the mixing chamber as a major source of erratic behaviour; (**b**) visual observation of a transitory regime while filling with coloured blue and yellow water.

**Figure 8 sensors-19-00566-f008:**
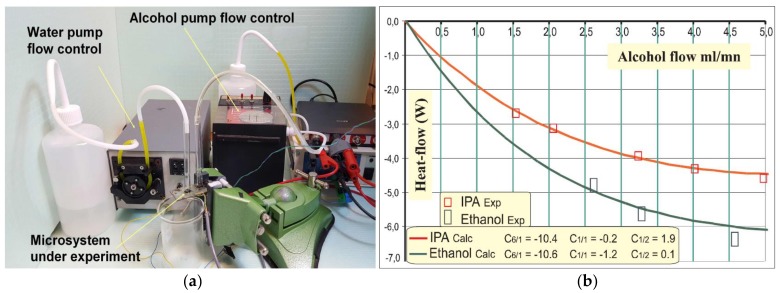
(**a**) Global view of the characterization system. (**b**) Compared values regarding measurements Φ_mes_ = V_out_/G·ξ and theoretical expected values calculated from Equation (4).

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
