# Peer review of "Enthalpy-Sensing Microsystem Effective in Continuous Flow†"

_sensors, 2019, doi:10.3390/s19030566_

Reviewer 1 Report

The english is a direct translation from French but the text is easy readable and I see no need for corrections. One question that I have - did you try to reduce air contnet in the water by using partial vacuum and if so was the air buble creation reduced?

Author Response

Dear Editor and the refrees,

Thank you for your careful reading of the first manuscript and your most useful comments. Please, find enclosed the duly revised version of the compuscript together with the relevant cover letter. In accordance with the comments of both reviewers, the paper has been amended along the whole text.

best regards

Reviewer 2 Report

This article describes two prototypes of a sensing system operating in continuous flow to measure enthalpies of mixing. The article looks good in general, and I find it suitable to be published in Sensors after considering a couple of issues. - Although the article is well written and clear, the structure is not so good. It is not 100% clear that the authors are explaining two different prototypes, or if the results belong to one or the other prototype, to cite a couple of examples. I think the article lacks a bit of structure, which can be easily improved. - Figure 1: What are the blue triangles? Are two different measurements from reference [11]. If so, some explanation should be provided without need of going to check the reference. Consider if there is a need to present this set of data if it's not going to be used. - Section Materials and Methods: As far as I understand, in this section the authors claim that the different materials (PET, PLA...) are ok with mixtures of water+alcohol, but then, afterwards, they have some problems with their sensor in this regard. Is this consistent? - What it "for analysis" quality? Could the authors provide a % purity? - Discussion: a comparative with other similar sensors would be interesting.

Author Response

(The authors gave the same response as above.)
